# Incremental Road Network Update Method with Trajectory Data and UAV Remote Sensing Imagery

**Jianxin Qin** [1,2,†], **Wenjie Yang** [1,2,†], **Tao Wu** [1,2,*], **Bin He** [1,2] **and Longgang Xiang** [3]

1    Hunan Key Laboratory of Geospatial Big Data Mining and Application, Hunan Normal University, Changsha 410081, China
2    School of Geographic Sciences, Hunan Normal University, Changsha 410081, China
3    State Key Laboratory of LIESMARS, Wuhan University, Wuhan 430079, China
*    Correspondence: blackender@hunnu.edu.cn
†    These authors contributed equally to this work.

**Abstract:** GPS trajectory and remote sensing data are crucial for updating urban road networks because they contain critical spatial and temporal information. Existing road network updating methods, whether trajectory-based (TB) or image-based (IB), do not integrate the characteristics of both types of data. This paper proposed and implemented an incremental update method for rapid road network checking and updating. A composite update framework for road networks is established, which integrates trajectory data and UAV remote sensing imagery. The research proposed utilizing connectivity between adjacent matched points to solve the problem of updating problematic road segments in networks based on the features of the Hidden Markov Model (HMM) map-matching method in identifying new road segments. Deep learning is used to update the local road network in conjunction with the flexible and high-precision characteristics of UAV remote sensing. Additionally, the proposed method is evaluated against two baseline methods through extensive experiments based on real-world trajectories and UAV remote sensing imagery. The results show that our method has higher extraction accuracy than the TB method and faster updates than the IB method.

**Keywords:** road network; trajectory data; UAV remote sensing imagery; deep learning

## 1. Introduction

Highly accurate and real-time road network data are essential for locally based urban applications, such as vehicle navigation, autonomous driving, and urban planning. The rapid growth of cities makes updating road networks challenging. For example, the government built many buildings and roads in new urban areas as urban boundaries expanded. Simultaneously, some roads in old urban areas are being reconstructed as part of urban renewal. As a result, rapid and accurate update of the road network has become a popular topic for research.

Current road network update methods mainly focus on TB [1,2] road network structure extraction and IB [3,4] road network structure extraction. There are two updating modes for TB methods: global updating and local incremental updating. The TB global update method reconstructs the whole road network by trajectories and replaces the original road network with the reconstructed road network. The local incremental update method identifies areas where the road network has changed, then updates the changed areas by using the trajectory data to generate the road network. TB road network update methods are typically speedy.

Nevertheless, the TB method produces less-accurate results than the IB method. TB approaches extract road network geometries from remote sensing images and update the road network within image coverage. These methods have better performance in accuracy but require a large amount of labor for visual interpretation, which is also time-consuming.

Additionally, the IB method's timeliness is inadequate because the latest satellite images are frequently unavailable. Neither TB nor IB methods can rapidly and accurately update road network data.

This paper proposes a novel incremental road network update method integrating GPS trajectory data and UAV remote sensing imagery. This method uses the characteristics of the HMM [5,6] map-matching method to identify new road segments and the connectivity of adjacent points in the matching process to identify problematic road segments. Deep learning technology is utilized to automatically extract the road segments from the digital orthophoto map (DOM) created by the UAV collecting picture data for the range area. Updating of new and problematic road segments is completed by reconstructing the road network topology. We make three main contributions compared to other methods, as follows:

1.  Composite framework for road network update: The framework integrates the features of vehicle trajectory and image data to rapidly and accurately update the road network.
2.  Problematic road segment identification and extraction algorithm: The algorithm utilizes the relationship between trajectory points and corresponding road segments during HMM map-matching to identify and extract problematic road segments.
3.  Method to integrate UAV remote sensing imagery and deep learning techniques: The method is based on the characteristics of these two techniques to quickly acquire images of update regions and automatically extract road segment boundaries from the images.

The article is organized as follows. Section 2 describes the related work for road network updating. Section 3 presents the methodological framework and details of this paper. Section 4 conducts the experimental study and comparative analysis of methods. Section 5 concludes the article and provides an outlook for the future.

## 2. Related Works

Existing methods for updating road networks can be divided into two types: (1) TB methods, characterized by rapid speed; and (2) IB methods, characterized by high accuracy.

### 2.1. TB Methods

TB road network updates currently employ two methodologies: global updating and local incremental updating. Global updating generates road networks by original trajectory and replaces the original road networks in the corresponding area [7]. Global updating methods include density-image skeleton extraction, point clustering, and incremental trajectory insertion. Density-image skeleton extraction extracts the road centerline from the density image [8]. By grouping the trajectory points into several clusters, the point clustering method creates a network by connecting the cluster nodes [9,10]. Incremental trajectory insertion constructs road networks by merging new GPS trajectories [11,12]. These techniques can quickly reconstruct the road network. However, because of geometric defects and GPS data sampling frequency, the produced road network has low accuracy, making it challenging to use in actual applications. Many researchers have recently focused on generating road networks from crowdsourced data [13]. Huang et al. proposed a method to integrate crowdsourced trajectories to achieve simultaneous geometric and semantic updates of road networks [14]. Luliang Tang et al. constructed a theoretical system for crowdsourced perception of high-precision maps based on traffic spatio–temporal big data [15]. However, it does not appear that the quality of the road networks created using these technologies has increased noticeably. In addition, Yali Li et al. proposed a global-scale method for road network generation by fusing taxi trajectory data and remote sensing images [16], which provides insight for our research.

Local incremental updating detects areas where the road network has changed by analyzing the relationship between trajectories and road network. This experiment reconstructs the road network in changing areas using trajectories. Point clustering methods [17–20] are frequently employed to generate local road networks, while spatial semantic information

of trajectories is also considered [21]. To update local road networks, researchers have used automatic road network change detection [22], trajectory identification [23,24], and geometric structure change detection [25]. Through the interaction between trajectory data and the road network, these approaches can efficiently identify locations where the road network has changed.

Nevertheless, accuracy needs to be increased because road network updates do not always match real situations. Ahmed [26] and Hashemi [27] et al. compared and analyzed various existing road network reconstruction methods. Meanwhile, the researchers were aware that using the strategies of collaborative iterative updating [28] and associating road segment nodes [1] can improve the quality of the road network However, the effect is not evident for complex areas of the road network. Yuxia Li et al. proposed a novel two-stage approach for inferring road networks from trajectory points and capturing road geometry with better accuracy [29].

### 2.2. IB Methods

The IB road network update method overlays the DOM with the original road network, assessing the geometric discrepancies between the road networks and the roads on the DOM, and modifying the new and problematic road segments. Abolfazl Abdollahi and Biswajeet Pradhan argued that methods for road network extraction from aerial images include decision tree (DT), k-nearest neighbor (KNN), and support vector machine (SVM), which need to consider spectral, geometrical, and texture information of images [30]; Favyen Bastani et al. proposed RoadTracer, a new method to automatically construct accurate road network maps from aerial images [31]. Hamid Reza Riahi Bakhtiari et al. proposed semi-automatic extraction of high-resolution images using edge detection, support vector machine, and mathematical morphology methods [32]. Abdollahi et al. used a deep learning approach for automatic extraction of road networks from remote sensing images; road segmentation and road extraction can be achieved [33–38] by deep learning technology. Rong Xiao et al. innovatively proposed a semi-supervised fully convolutional neural network to extract road segment information efficiently at a low cost [39]. Lin Gao et al. proposed a method based on a deep residual convolutional neural network to extract road information from complex scenes [40]. Jiang Xin et al. proposed a new method to extract the road network from remote sensing images using a DenseUNet model with few parameters and robust characteristics [41]. Ziyi Chen et al. proposed a reconstruction-bias U-Net for road extraction from remote sensing images and achieved improved performance [42]. Calimanut-Ionut Cira et al. proposed a novel framework based on convolutional neural networks (CNNs) to classify secondary roads in high-resolution aerial orthoimages [43]. Naveen Chandra et al. explored road extraction using high-resolution images by a cognitive analysis model [44]. Jiguang Dai et al. proposed a lane-level road network extraction method based on high resolution images to improve the automation of road extraction by setting different scenes [45]. The IB method can improve road network accuracy by continuously optimizing the algorithm, but it requires powerful computing and long processing times.

These studies have solved many problems. However, the accuracy of the TB method is poor, and the IB method takes a long time. Existing methods cannot simultaneously achieve rapid and accurate updating of road networks. Therefore, this paper proposes an incremental update method of the road network with trajectory data and UAV remote sensing imagery. The method proposes a composite road network update framework that combines trajectory and UAV image data and identifies problematic road segments by using the connectivity between adjacent matching points in the process of map-matching. The method also introduces a deep learning technique to identify road boundaries. The proposed method combines the advantages of TB and IB methods to achieve rapid and accurate updating of the road network.

## 3. Methodology

As shown in Figure 1, this study proposes a novel method to incrementally update road network data from trajectories and UAV images on the local scale, including data pre-processing, update region identification, update region image acquisition, and local road network updates. A three-phase update strategy is designed to detect potential target locations and extract corresponding road segments. In the first phase, an HMM-based detection algorithm integrates trajectories with original road network data for identifying potential update targets. In the second phase, a UAV captures images of potential targets in the real world and generates a DOM. In the third phase, road segments are extracted from images by deep learning techniques [46] for updating road network data on the local scale. Such a strategy benefits road network data updating accuracy and efficiency with the idea of local renewal.

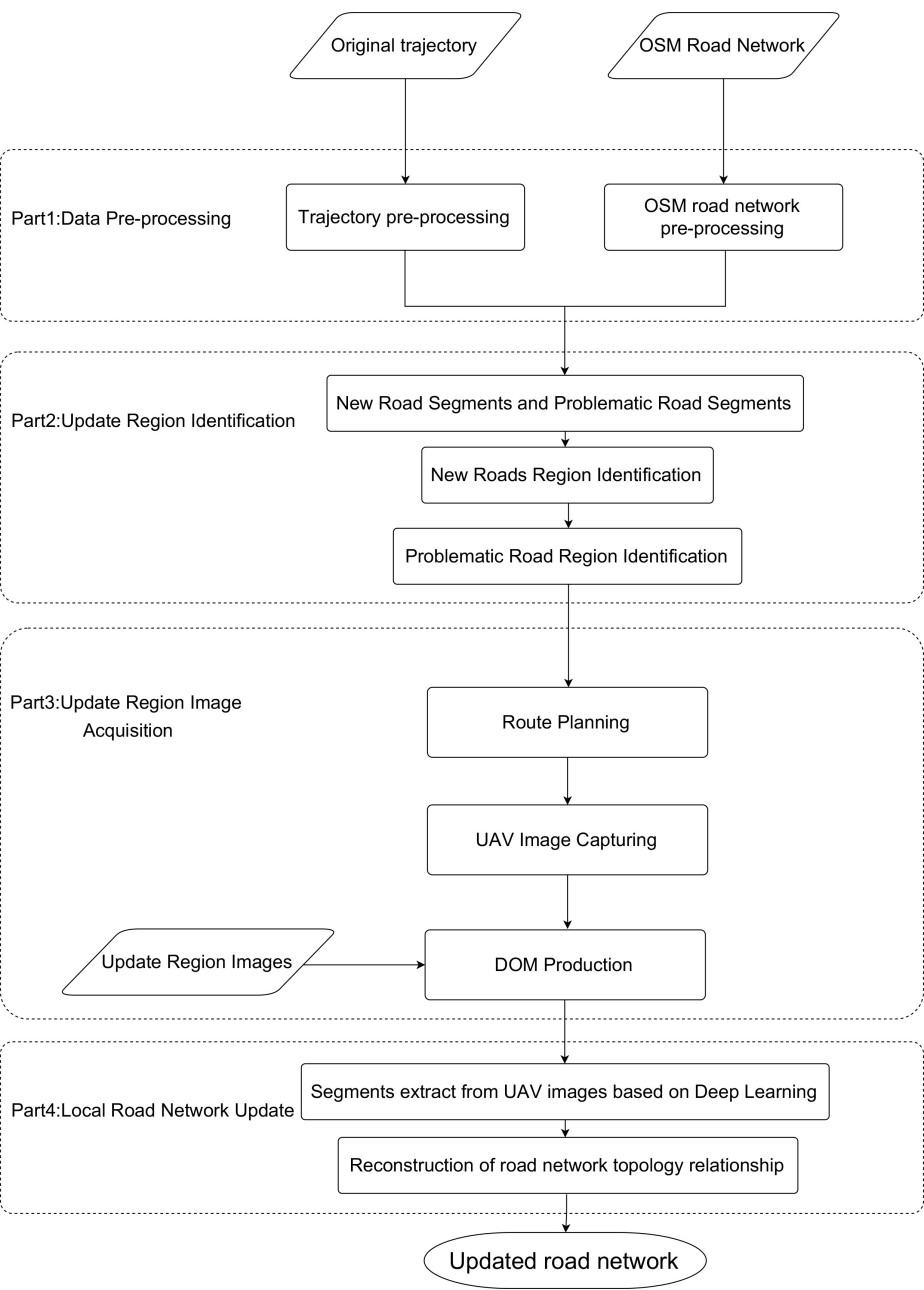

**Figure 1.** Workflow of the proposed method.

### 3.1. Data Pre-Processing

Obtaining road network data in shapefile format from the OSM (OpenStreetMap) website causes loss of some of the node and direction information of the road segment. The road network data required in this paper is a directed graph $G$ containing nodes and edges. Therefore, processing the road network in shapefile format is necessary to obtain the directed graph $G = (V, E)$. $V = (v_1, v_2, \ldots, v_n)$ represents the node of the road, and $n$ represents the index of the node; $E = (e_1, e_2, \ldots, e_m)$ represents the road segments, and $m$ represents the index of the road segments. Each road segment contains five attributes: identification code ($id$), starting node ($source$), ending node ($target$), design speed (km/h), and segment length ($weight$).

The raw trajectory data need to be processed as shown in Figure 2b, including the five attributes of the trajectory point ID ($uuid$), trajectory ID ($track\_id$), the acquisition time of the trajectory point ($log\_time$), the moving object ID ($car\_id$), and the speed ($v$). The original vehicle tracks are tagged with license plates and collected chronologically. As shown in Figure 2a, the raw vehicle trajectories only contain trajectory point acquisition time ($log\_time$), longitude ($longitude$), and latitude ($latitude$). The longitude ($longitude$) and latitude ($latitude$) can calculate the Euclidean distance between different trajectory points and the velocity ($v$) of each trajectory point. To ensure the quality of each trajectory, the method divided trajectories into sub-trajectories when time or data are missed. Each trajectory's ID ($track\_id$) is assigned a value starting from 1. Each trajectory's point ID ($uuid$) is named in ascending order starting from 1. The raw trajectory contains the license plate number ($car\_id$) and trajectory point acquisition times ($log\_time$). The data processing process includes the following steps:

1. Delete the trajectory points outside the study area.
2. Obtain the distance (dist) between the different trajectory points based on longitude and latitude. Then, the velocity ($v$) of the trajectory points can be obtained by the distance ($dist$) and the acquisition time ($log\_time$).
3. Obtain the license plate number ($car\_id$) from the file name of the original data.
4. Eliminate the noisy points with speed less than 5 km/h or greater than 120 km/h.
5. Number the trajectories and get the trajectory ID ($track\_id$).
6. Generate the trajectory point ID ($uuid$) for each trajectory in increasing order starting from 1.
7. Merge the trajectory file corresponding to each trajectory.
8. Project the trajectory file to UTM. The final trajectory data shown in Figure 2b can be obtained.

| | A | B | C |
|---|---|---|---|
| | log_time | longitude | latitude |
| 1 | | | |
| 2 | 2021/1/1 0:00 | 118131328 | 32060475 |
| 3 | 2021/1/1 0:03 | 118103631 | 32054210 |
| 4 | 2021/1/1 0:05 | 118073031 | 32045953 |
| 5 | 2021/1/1 0:07 | 118055741 | 32038928 |
| 6 | 2021/1/1 0:10 | 118020171 | 32026938 |
| 7 | 2021/1/1 0:27 | 117798540 | 31984230 |
| 8 | 2021/1/1 0:32 | 117744928 | 31974261 |
| 9 | 2021/1/1 0:33 | 117733336 | 31966256 |

(a)

| uuid | track_id | log_time | car_id | v |
|---|---|---|---|---|
| 1 | 20 | 2021-01-01 15:10:54 | A01754 | 43 |
| 2 | 20 | 2021-01-01 15:11:24 | A01754 | 40 |
| 3 | 20 | 2021-01-01 15:11:54 | A01754 | 8 |
| 4 | 20 | 2021-01-01 15:12:54 | A01754 | 21 |
| 5 | 20 | 2021-01-01 15:13:24 | A01754 | 28 |
| 6 | 20 | 2021-01-01 15:13:54 | A01754 | 39 |
| 7 | 20 | 2021-01-01 15:14:24 | A01754 | 24 |
| 8 | 20 | 2021-01-01 15:14:54 | A01754 | 28 |
| 9 | 20 | 2021-01-01 15:15:24 | A01754 | 35 |

(b)

**Figure 2.** Trajectory data pre-processing: (**a**) raw trajectory data; (**b**) pre-processed trajectory data.

### 3.2. Update Region Identification

The purpose of identifying the update regions was to narrow the scope of road network updating. The proposed method can identify two kinds of update regions: new road segments and problematic road segments. Appropriate identification methods can be designed based on the contour of the trajectory and its geometric characteristics reflected in the road network. For example, vehicle trajectories corresponding to new road segments usually regularly keep a certain distance from the road segments. Most of these trajectory

points do not have corresponding projected road segments. In the HMM-based map-matching process, the direction of the trajectory is often opposite to the direction of its corresponding problematic road segment. Therefore, the connectivity between adjacent matching points can identify the problematic road segments.

### 3.2.1. New Road Segments and Problematic Road Segments

"New road segments" refers primarily to new roads constructed in urban areas. In geometric form, the new road segments intersect or border other road segments. The trajectory points generated by vehicles driving on the new road segments usually keep a certain distance from other road segments, and their geometric characteristics are regular and sequential. As shown in Figure 3, the sequence $p_9 \rightarrow p_{10} \rightarrow p_{11} \rightarrow p_{12}$ inside region A4 refers to a vehicle driving on a new road segment.

Problematic road segments in this paper refer to those that are erroneously updated in their direction or are delayed in updating.

1.  The original road segments were not updated on the road network system in time after reconstruction, including the situation shown in Figure 4a. As shown in Figure 3, there is a sequence of trajectory points $p_{18} \rightarrow p_{19} \rightarrow p_{20}$ in region A1. The original road segment is a one-way road segment, which was not updated in time after its expansion into a two-way road segment.
2.  The segment direction was updated incorrectly, which means that the road segment's direction was updated to the road network system incorrectly, including the two cases shown in Figure 4b. One is the error of the one-way road segment that goes opposite to the actual situation, as shown in Figure 3 for the sequence of trajectory points $p_{15} \rightarrow p_{16} \rightarrow p_{17}$ in region A2. The other is the error of two-way road segments that have been updated into only one direction, as shown in the sequence of trajectory points $p_1 \rightarrow p_2 \rightarrow p_3 \rightarrow p_4$ in region A3 in Figure 3. These types of errors cannot be identified by visual interpretation.

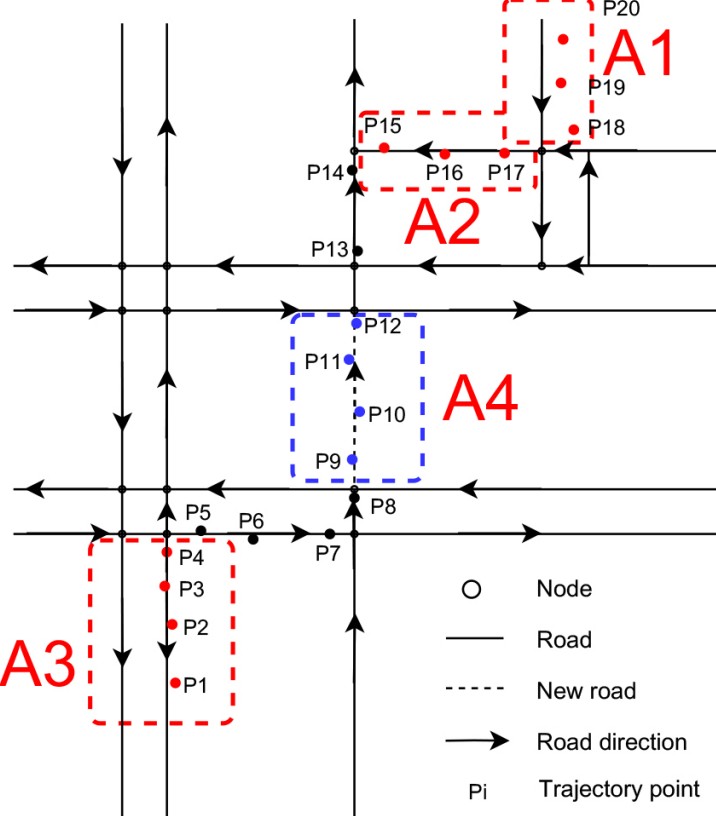

**Figure 3.** Update region identification. A1, A2, and A3 represent the problematic road segment regions. A4 represents the new road segment regions.

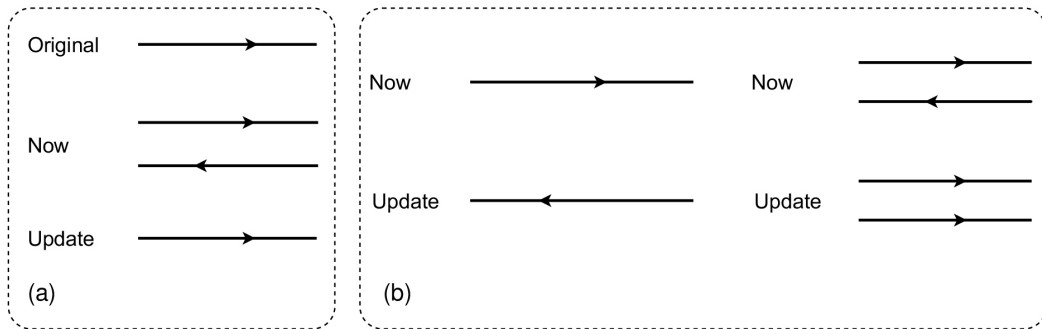

**Figure 4.** Categories of problematic road segments: (**a**) road segments that are not updated promptly; (**b**) road segments that are updated incorrectly with respect to their direction.

### 3.2.2. New Roads Region Identification

Algorithm 1, named *Identify new road segments*, identifies sequences within trajectories that travel through new roads based on HMM-based map-matching. The HMM-based map-matching method consists of observation and transmission probability models. Formally, the equation of observation probability has been defined as $N\left(c_i^j\right)$ of $c_i^j$ with respect to $p_i$:

$$N\left(c_i^j\right) = \frac{1}{\sqrt{2\pi}\sigma}e^{-\frac{\left(x_i^j-\mu\right)^2}{2\sigma^2}} \tag{1}$$

where $x_i^j = \mathrm{dist}\left(c_i^j, \quad p_i\right)$ is the distance between $p_i$ and its project points $c_i$. Based on empirical analysis, the standard deviation for the normal distribution used in this study is 30 m.

The transmission probability determines transit between the candidate positions on the road network by applying the following formula:

$$V\left(c_{i-1}^t \to c_i^S\right) = \frac{d_{i-1\to i}}{w_{(i-1,t)\to(i,s)}} \tag{2}$$

where $d_{i-1\to i} = \mathrm{dist}(p_i, p_{i-1})$ is the Euclidean distance between $p_i$ and $p_{i-1}$, and $W(i-1,t) \to (i,s)$ ) is the length of the shortest path from $c_{i-1}^t$ to $c_i^t$.

The observation probability model of the HMM map-matching method can be used to identify the trajectories of vehicles that travel on the new road segments. The observation probability model calculates the observation probability between a trajectory point and its corresponding projection point. A circle $O$ with each trajectory point was drawn in the trajectory. The trajectory point is the center of the circle $O$. The maximum geometric error $r$ is the radius of the circle $O$. When the circle $O$ intersects or is tangent to the road segment $E_i$, the point $C_i$ with the closest distance from the trajectory point to the road segment $E_i$ is the projection point of the trajectory point on the corresponding road segment. The vehicle's trajectory is usually on the road network during its travel. The observation probability model can project the trajectory points onto the corresponding road segments. A portion of the vehicle trajectories could not be matched with the corresponding road segments. This part of the trajectory was considered to be generated by the vehicle driving on the new road segments. Our pseudo-code is shown in Algorithm 1; it is processed as follows:

1.  Input the road network $G$, the trajectory $T$, and the number of trajectory points $N$.
2.  Initialize dictionaries $C$, $L$, and $S$.
3.  Iterate over the trajectory points $p\_i$ in the trajectory $T$. If there are projection points for $p\_i$, assign $S$ to $p\_i$. Then, add $S$ to $C$, and empty $S$ again.
4.  Remove the trajectory points in the dictionary $C$ from the original trajectory $T$. The remaining trajectory points $L$ are the vehicle trajectories corresponding to the new road segments.

The trajectory points $p_9$, $p_{10}$, $p_{11}$, and $p_{12}$ shown in region A4 in Figure 3 are clearly generated by the vehicle driving on the new road segment. The trajectory $T(p_1 \rightarrow p_2 \rightarrow \ldots \rightarrow p_{20})$ as the input data of Algorithm 1 can output the trajectory points $L(p_9, p_{10}, p_{11}, p_{12})$.

---

**Algorithm 1** Identify new road segments.

---

**Input:** Road network $G = (E, V)$; trajectories $T : p\_1 \rightarrow p\_2 \rightarrow p\_3 \rightarrow \ldots \rightarrow p\_n$; $N.//N$
   represents the number of points in $T$
**Output:** The new road segment's points $L : l\_1 \rightarrow l\_2 \rightarrow l\_3 \rightarrow \ldots \rightarrow l\_n, 1 \leq i \leq n, l\_i \in T$.
  1: Initialize $C$, $L$, and $S$ as an empty dictionary
  2: **for** $i$ .. $N$ **do**
  3:     $S$=GetProjectPoints$(pi, G, r);//$ pi with radius r
  4:     $C$.add$(S)$
  5: **end for**
  6: $L = T \backslash C$
  7: **return** $L$

---

### 3.2.3. Problematic Road Region Identification

This study designs Algorithm 2 to identify the problematic segments. It has been divided into four parts: input, output, initialization steps, and the algorithm's core. The algorithm is named *Identify and extract problematic road segments*. The input of Algorithm 2 includes the road network $G = (E, V)$ and the set of trajectory points $C$. The output of Algorithm 2 includes the problem segments $L(l_1, l_2, \ldots, l_n)$. It is initialized with lines 1–4 in Algorithm 2, including:

1.  Obtain the projection points of the trajectory point and calculate the observation probability from the trajectory point to its projection points.
2.  Calculate the transmission probabilities between adjacent projection points.
3.  Find the set $S$ of the sequence of trajectory points corresponding to the maximum probability path $S : s_1 \rightarrow s_2 \rightarrow s_3 \rightarrow \ldots \rightarrow s_n$.
4.  Set the probability threshold $SMALL\_PROBABILITY = 0.00000001$ and initialize $N$, $M$, and $L$ as the empty dictionary. $N$ represents the set of unconnected points, $M$ represents the set of connected points, and $L$ represents the set of problematic segments.

The core part of the algorithm is lines 5–38 in Algorithm 2, and its design idea is represented in Figure 5. The trajectory points in $S$ are iterated through initially. The value *current_point* represents the current point; *next_point* represents the next point. When the transmission probability between *current_point* and *next_point* is greater than the set threshold $SMALL\_PROBABILITY$, *current_point* is connected to *next_point*. The relationship between adjacent matching points in $S$ can be divided into the following four cases:

1.  If *current_point* is connected to *next_point*, add *current_point* to $M$, as shown in steps 0-2 in Figure 5. If *next_point* is also the last point of $S$, add it to $M$.
2.  If *current_point* does not connected with *next_point*, and *current_point* is the first point of $S$, then *current_point* is added to $N$, as shown in steps 0-1-4 in Figure 5. If *next_point* is also the last point of $S$, add it to $N$.
3.  If all three conditions are satisfied: *current_point* does not connect with *next_point*, *current_point* is not the first point of S, and *current_point* is connected with its previous point; then, *current_point* is added to $M$, as shown in steps 0-1-3-6 in Figure 5. If *next_point* is also the last point of $S$, it is added to $N$.
4.  If all three conditions are satisfied: *current_point* does not connect with *next_point*, *current_point* is not the first point of $S$, and *current_point* is not connected with its previous point; then, *current_point* is added to $N$, as shown in Figure 5, steps 0-1-3-5. If *next_point* is also the last point of $S$, it is added to $N$.

---

**Algorithm 2** Identify and extract problematic road segments.

---

**Input:** Road network $G = (E, V)$; Trajectory points $C$.

**Output:** The problematic road segments $L : l_1, l_2, \ldots, l_n$

  1: HMM_based get project points and calculate its observation;

  2: HMM_based calculate transmission between adjacent project points;

  3: Initialize $N$, $M$, and $L$ as an empty dictionary, SMALL_PROBABILITY = 0.00000001;
     //N representd the set of unconnected points, M represents the set of connected points, L represents the set of problematic road segments.

  4: Obtain the set of project points with the maximum probability path and call it match_point_list $S : s_1 \rightarrow s_2 \rightarrow s_3 \rightarrow \ldots \rightarrow s_n$; obtain the number of points in $S$, and call it $X$.//the points of S are not necessary connected to each other; $X$ represents the number of points in $S$

  5: **for** $i \, .. \, X$ **do**

  6:     $current\_point = s[i]$;// current_point represents the current point

  7:     $next\_point = s[i + 1]$;// next_point represents the next point

  8:     $transmission\_probability = g[current\_point][next\_point]['transmission\_probability']$;

  9:  //transmission_probability represents the connection between current_point and next_point, if transmission_probability=SMALL_PROBABILITY, it means current_point and next_point are not connected; otherwise are connected

 10:     **if** $transmission\_probability = SMALL\_PROBABILITY$ **then**

 11:       **if** $current\_point$ is not the first point of $S$ **then**

 12:         **if** $current\_point$ is not connected with its previous point **then**

 13:           $N$ add.(current_point)

 14:           delete the first point of $S$

 15:           **if** next point is the last point **then**

 16:             $N$ add.(next_point)

 17:           **end if**

 18:         **else**

 19:           $M$ add.(current_point)

 20:           delete the first point of $S$

 21:           **if** next point is the last point **then**

 22:             $N$ add.(next_point)

 23:           **end if**

 24:         **end if**

 25:       **else**

 26:         $N$ add.(current_point)

 27:         delete the first point of $S$

 28:         **if** next point is the last point **then**

 29:           $N$ add.(next_point)

 30:         **end if**

 31:       **end if**

 32:     **else**

 33:       $M$ add.(current_point) delete the first point of $S$

 34:       **if** next point is the last point **then**

 35:         $M$ add.(next_point)

 36:       **end if**

 37:     **end if**

 38: **end for**

 39: Search the road $ID$ corresponding to the point in $N$, and add it to $L$

 40: Remove duplicate segments in $L$

 41: **return** $L$;

---

The set of disconnected points $N$ can be obtained by traversing every trajectory point in $S$. Each unconnected point in $N$ has the corresponding associated road segments $l_i$; output these associated road segments and remove the duplicate road segments to get the final extracted problematic road segments $L(l_1, l_2, \ldots, l_n)$.

For example, as Figure 3 shows, regions A1, A2, and A3 correspond to different problematic road segments, which can be extracted by Algorithm 2. The problematic segment indicated by A1 is the one-way segment that has not been promptly updated on the road network database despite being converted to the two-way segment, as shown in Figure 4a. The extraction process combined steps 0-1-4 and steps 0-1-3-5 in Figure 5. The problematic segment indicated by A2 is the one-way segment direction update error, as shown in Figure 4b. The extraction process was the combination of steps 0-1-4, steps 0-1-3-5, and steps 0-1-3-6 in Figure 5. The problematic segment indicated by A3 is the two-way segment direction update error, as shown in Figure 4b. The extraction process combined steps 0-1-4 and steps 0-1-3-5 in Figure 5.

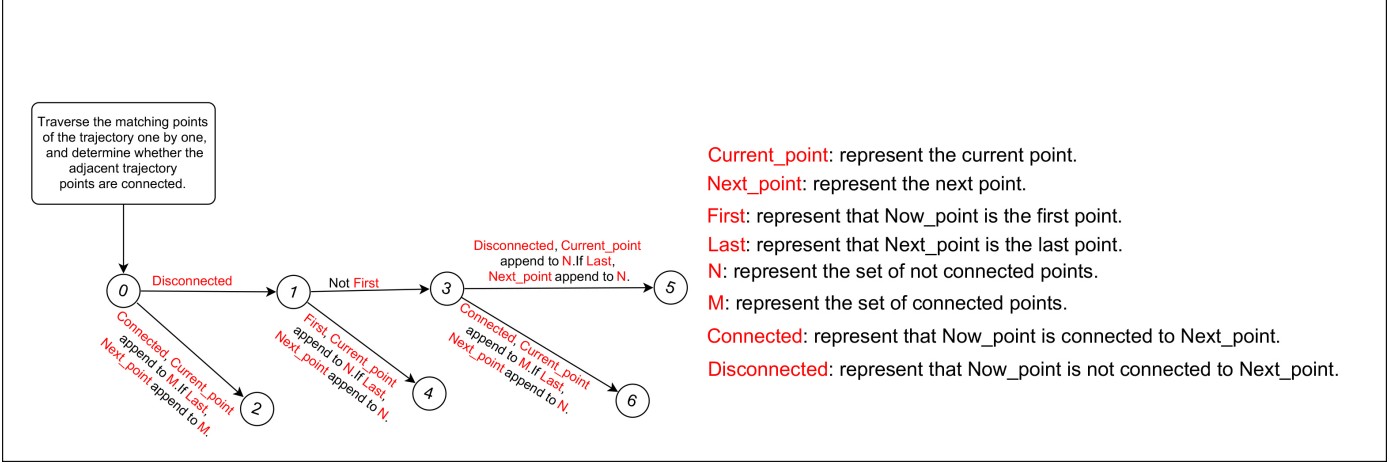

**Figure 5.** Algorithm of problematic road segment region extraction. The arrows indicate the execution process of the algorithm. The nodes indicate the execution steps of the algorithm.

### 3.3. UAV Image Acquisition of Update Regions

This study uses a UAV to acquire images of update regions. UAV remote sensing technology is a new type of surveying and mapping technology that has emerged in recent years with the development of UAV technology. UAV technology has the characteristics of low cost, high efficiency, and high resolution compared with the technology of acquiring images through satellites. The acquired images can be stitched into a DOM after processing.

This research uses a UAV with RTK positioning capability and a high-resolution digital camera to acquire images of the update regions. The RTK positioning technique enables the POS accuracy of the images to be 3 cm. To improve operational efficiency, the high-resolution digital camera allows the UAV to obtain images of the corresponding regions at higher flight altitudes. The operation route is shown in Figure 6a; the UAV operates only in the update regions. The acquired images of the update regions are shown in Figure 6b; the images are arranged regularly on both sides of the centerline of the update regions.

As shown in Figure 7, the acquired images need to be processed by the following steps:

1. Calculate the outer orientation elements of images by aerial triangulation.
2. Use the DEM model to remove the distortions of images due to the irregular terrain.
3. Splice the corrected images.
4. Adjust the inlaid line.
5. Unify the color and light of all images.
6. Export the DOM of regions.

**Figure 6.** Route and images: (**a**) UAV route in update regions and (**b**) images acquired by UAV (DJI Terra [47]).

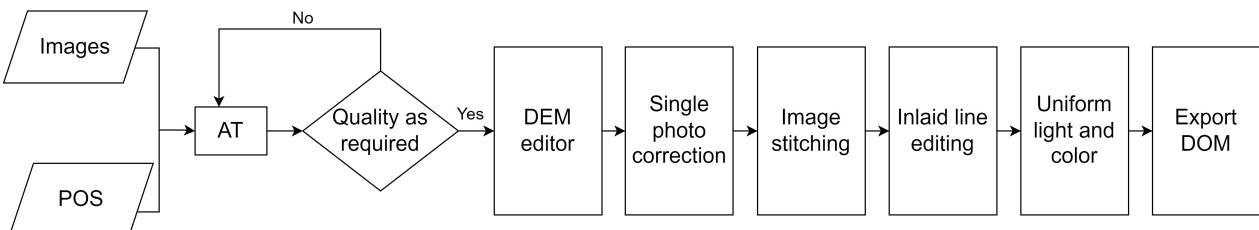

**Figure 7.** Production of DOM.

### 3.4. Local Road Network Update

Road network updating is based on a composite update strategy, including deep learning-based road segment extraction from UAV images and road network updating. This paper uses a convolutional neural network (CNN) to extract the boundaries of road segments from the images. The directional attributes of the trajectories are added to improve the efficiency and accuracy of road network updating.

### 3.4.1. Deep Learning-Based UAV Image Road Segment Extraction

A combination of U-Net [48] and ResNet [49] was employed to identify the boundaries of road segments on the DOM, as shown in Figure 8. U-Net is an end-to-end deep learning model typically characterized by a U-shaped symmetric structure. The first half of the U-Net network acts as feature extraction, and the second half is upsampling, usually called the encoder–decoder [50]. The encoding part of the U-Net network consists of four convolutional layers, mainly used to extract high-level semantic information from the images. The decoding part contains four upsampling layers, and the high-level semantic feature map obtained from the encoding part is restored to the resolution of the original image by transposition convolution. U-Net uses skip connections in the same stage as the upsampling process instead of direct convolution and transposition convolution of high-level semantic features. Skip connections ensure that the final recovered feature map incorporates more low-level spatial information and integrates the spatial information of different scales, making the segmentation map recover more fine information, such as edges. U-Net is an excellent semantic segmentation model but is prone to gradient disappearance during the encoding process. The information lost in different layers of the encoding is preserved, and the convergence of gradient descent is accelerated by introducing ResNet in the coding process. Higher-level feature images can be obtained. The main structure of the model is shown in Figure 8.

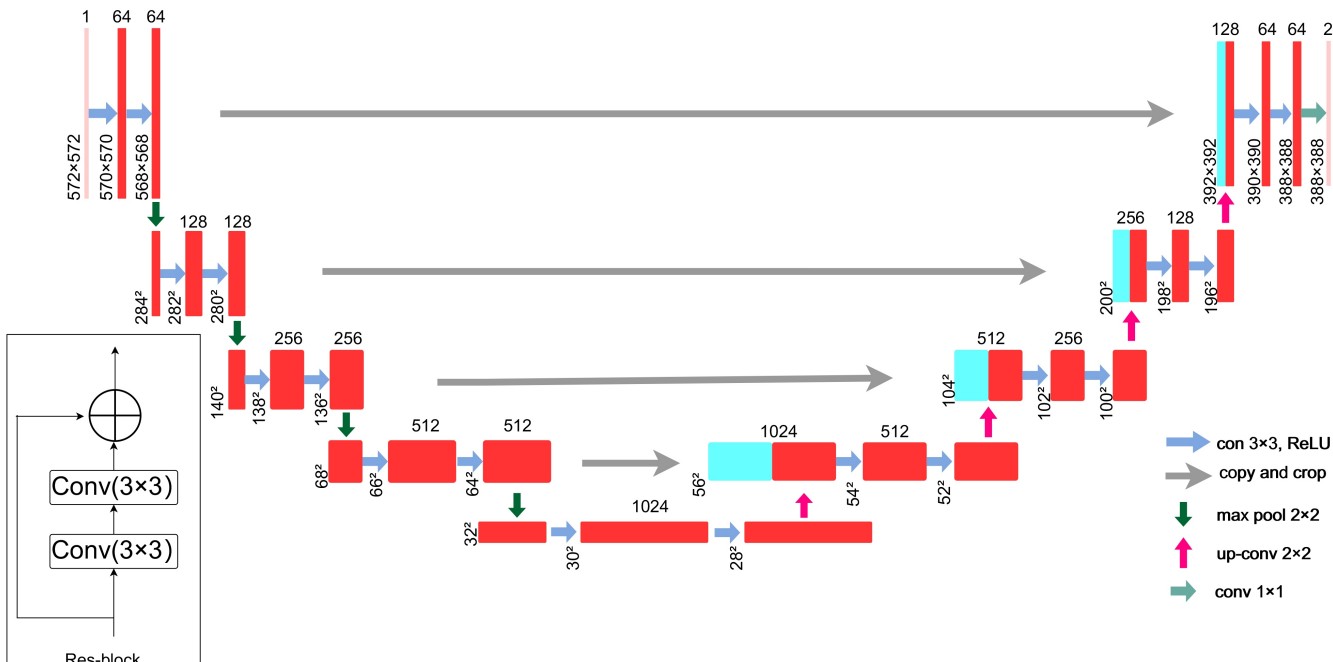

**Figure 8.** A deep learning framework combining U-net and ResNet.

### 3.4.2. Road Network Update

The method uses a composite update strategy that combines these three spatio–temporal data sources, including trajectory, road network, and UAV images, to update the road network. As shown in Figure 9, our strategy was updated for two road segments separately: new and problematic road segments. The new road segments were not updated

in the road network database, and their geometry was unsure. It was uncertain whether the geometry of the problematic road segments had changed.

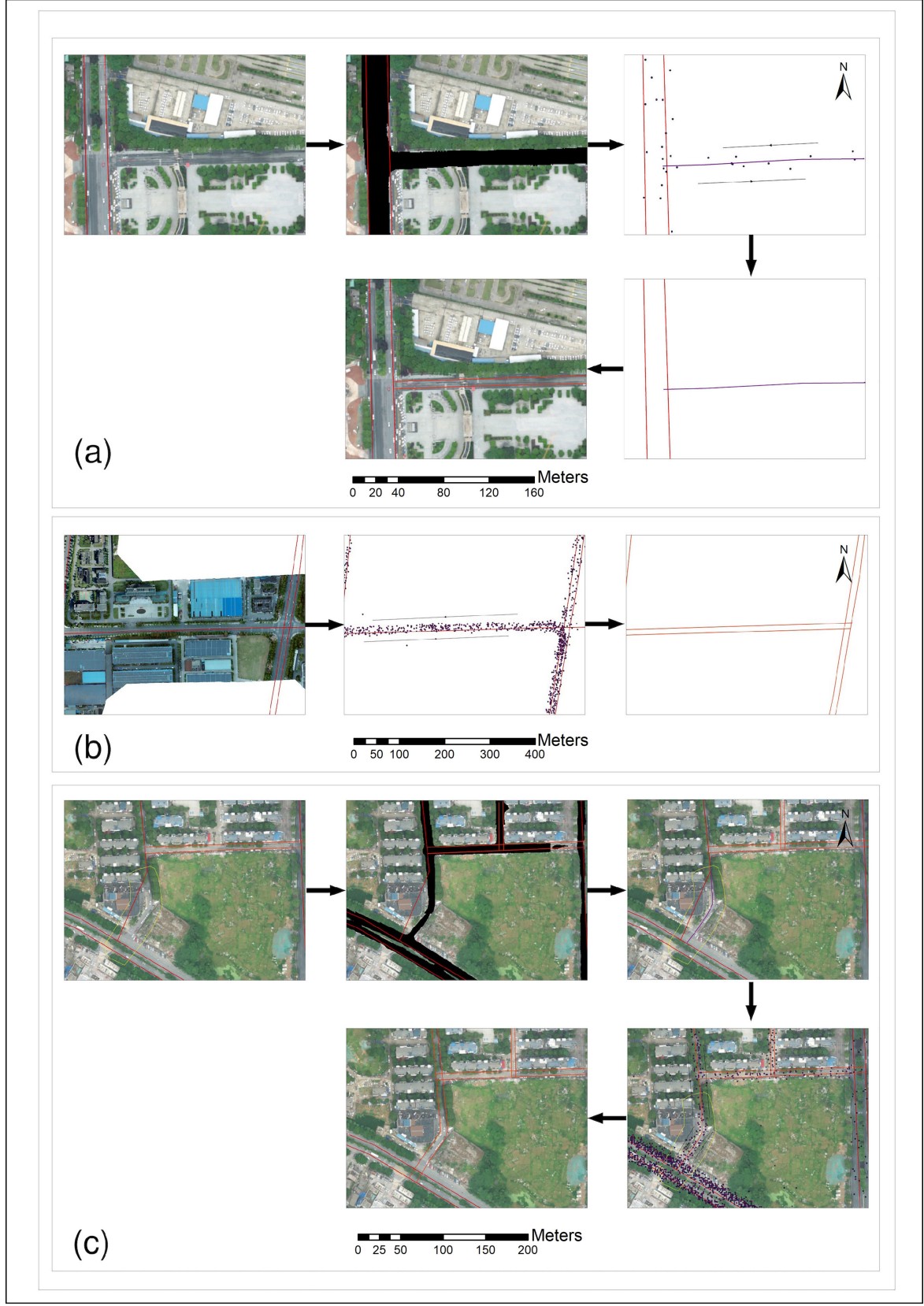

**Figure 9.** Update process for road segment: (**a**) the new road segments; (**b**) the problematic segments where the geometry has not changed; and (**c**) the problematic segments where the geometry has changed.

The CNN technique for road extraction from the DOM of its corresponding region is applied for the new road segments. As shown in Figure 10, the output of the deep learning for road extraction is a raster image. The extracted road is white in the image. The other features are represented in black. The raster image is updated to the actual road segment through plural filtering, refinement, raster to the surface, surface to the line, and smoothing operations.

The research adds the directional attributes of the trajectory data for the road network update. Directional information is considered an essential feature of road segments when updating a road network, which makes up for the deficiencies of convolutional neural network technology and realizes the updating of two-way road segments. By combining the directional attributes of the trajectory, if the road segment corresponds to two directions, it will be updated as a two-way road. If the road segments correspond to only one direction, they will be updated as one-way road segments. As shown in Figure 9, the updating process of a road network can be discussed in three cases:

1. As shown in Figure 9a, the directional attribute of the trajectory has been combined to update the road network, mainly to estimate the new road segment as a one-way or two-way road segment.
2. As shown in Figure 9b, the road network has been updated using the directional property of the trajectory, specifically to identify whether the problematic road segment is a one-way or two-way road segment.
3. As shown in Figure 9c, the direction attributes of the trajectory have been combined to update the problematic road segments with changed geometries and to estimate whether they are one-way or two-way segments.

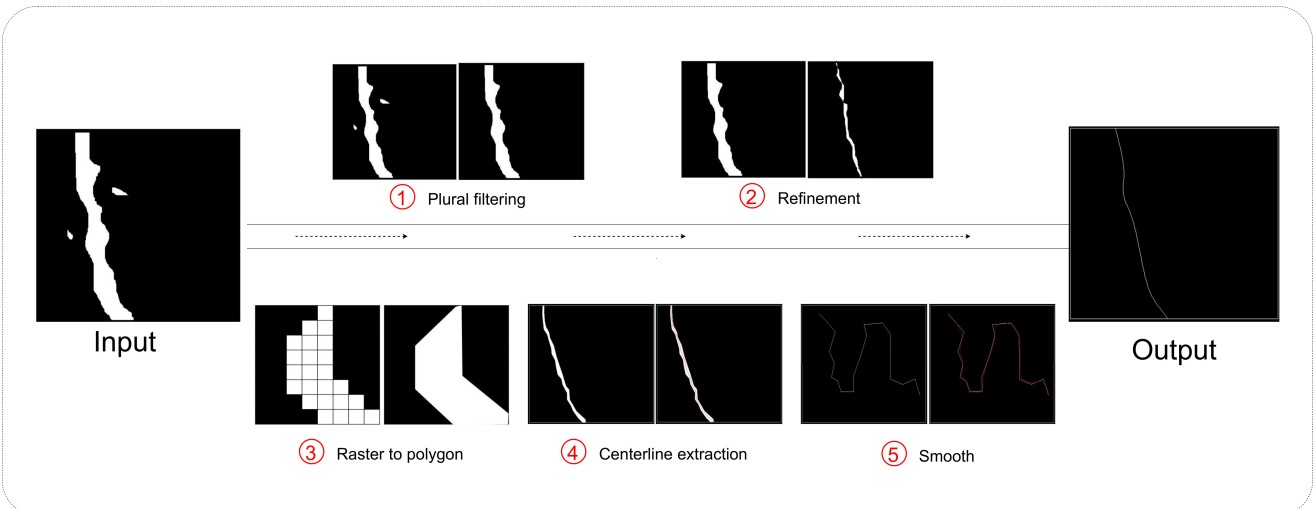

**Figure 10.** Post-processing of the CNN output.

## 4. Experiment

This section conducted real-world road network data experiments to validate the proposed method. Firstly, the basic characteristics of road network, vehicle trajectory, and UAV image data were introduced. Secondly, indicators were introduced to evaluate the experimental results, including precision, recall, and F-score. Finally, the experiments were conducted to compare the proposed method with other methods. The hardware configuration of the computer used in the experiment was as follows: Nvidia RTX 2060 GPU, 32G of RAM, and an Intel 8700K CPU.

### 4.1. Data Introduction

Most of the new urban areas and part of the old urban areas in the Yuelu District of Changsha City were selected as the experimental areas. As shown in Figure 11a, the study area is about 36 square kilometers. The experimental data comprised the road network,

trajectories, and UAV images. As a key development area of Changsha, Yuelu district has been actively constructing and remodeling its urban area this year to accommodate evolving travel requirements. Some roads in the old city have been renovated to meet the travel demands of people. These conditions provided an appropriate basis for our study.

The road network data was downloaded from the OSM website and was updated to 2019. The trajectory data were obtained from 4817 GPS-equipped floating cars traveling within the research region, with a total of 33,179 trajectories, as shown in Table 1. The time range for each trajectory was 0:00–24:00 and included but was not limited to cabs. The trajectories were from 1 January 2021 to 7 January 2021, and the sampling frequency ranged from 30 s to 2 min. The processed trajectory data are shown in Figure 11a. The processing of the raw trajectory consisted of the following steps:

1. Extract the trajectory data within the research area according to the research range longitude 112°50′29″ to 112°57′33″ and latitude 28°06′21″ to 28°14′20″.
2. Calculate the speed and distance of the trajectory points according to the different positions and time differences between adjacent trajectory points.
3. Slice sub-trajectories according to the time and distance thresholds between adjacent trajectory points.
4. Renumber the sliced trajectories by obtaining the attribute of moving object ID from trajectory data.
5. Number the trajectory points of each trajectory in ascending order starting from 1.
6. Merge all trajectory data into one file and convert it to UTM-49N projection.

A DJI Phantom 4 RTK UAV flying at an altitude of 180 m captured the image data with a heading overlap of 80% and a collateral direction overlap of 70%. As shown in Figure 11c, the ground resolution of the images is about 5 cm. As shown in Figure 11b, the images were stitched into a regional DOM after data processing.

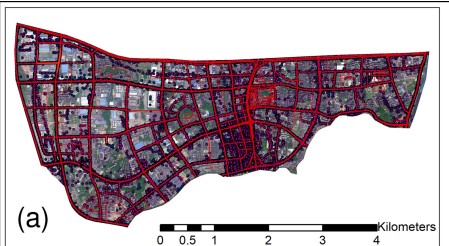 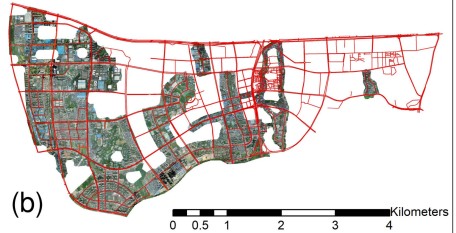 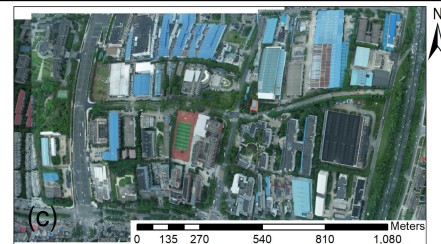

**Figure 11.** Data introduction of the experiment area: (**a**) trajectory and road network data; (**b**) DOM; and (**c**) images acquired by UAV.

**Table 1.** Trajectory data description.

| City | Time | Vehicles | Trajectories | Sampling Rate |
| --- | --- | --- | --- | --- |
| ChangSha | 2021-01-01 to 2021-01-07 | 4817 | 33,179 | 30 s to 2 min |

*4.2. Evaluation Indicators*

To quantitatively evaluate the performance of the proposed method, *precesion*, *recall*, and *F-score* [51] were used.

$$Precision(P) = length(TP)/length(TP + FP) \tag{3}$$

$$Recall(R) = length(TP)/length(TP + FN) \tag{4}$$

$$F - score = (2 * P * R)/((P + R)) \tag{5}$$

where $TP$ denotes the correctly predicted road segments, and $length(TP)$ denotes the length of the correctly predicted road segments. $FP$ denotes the incorrectly predicted road

segments, and $length(FP)$ denotes the length of the incorrectly predicted road segments. $FN$ denotes the road segments that were not predicted, and $length(FN)$ denotes the length of the road segments that were not predicted.

### 4.3. Experiment Result

In this paper, a composite update strategy is used to update two types of roads: new road segments and problem road segments. The experiment results are compared with the TB method and IB method.

### 4.3.1. Comparison with the TB Method

The method has obvious advantages for the update of new road segments. Figure 12a shows the update of the new road segments in the research region based on the latest satellite images of the regions. We find that the updated results match well with the actual situation by combining the updated area with the DOM for overlay analysis (as shown in A–E). The method of Deng et al. identifies most road segments in the study area, as shown in Figure 12b. However, it does not achieve better results for the sparse region of trajectories shown in the locally enlarged figure A. Meanwhile, the method adopts a T-shaped [22] strategy in the road network trajectory generation process, which leads to some added road segments inconsistent with the actual situation. For the Wu et al. approach, the newly added T-shaped road segments are not well-constructed due to the assumption that the mismatch points in the problem area are all located on a new road segment. Areas A, B, and E in Figure 12c have multiple new road segments that should be updated. In addition, neither the methods of Deng nor Wu can update bidirectional road segments.

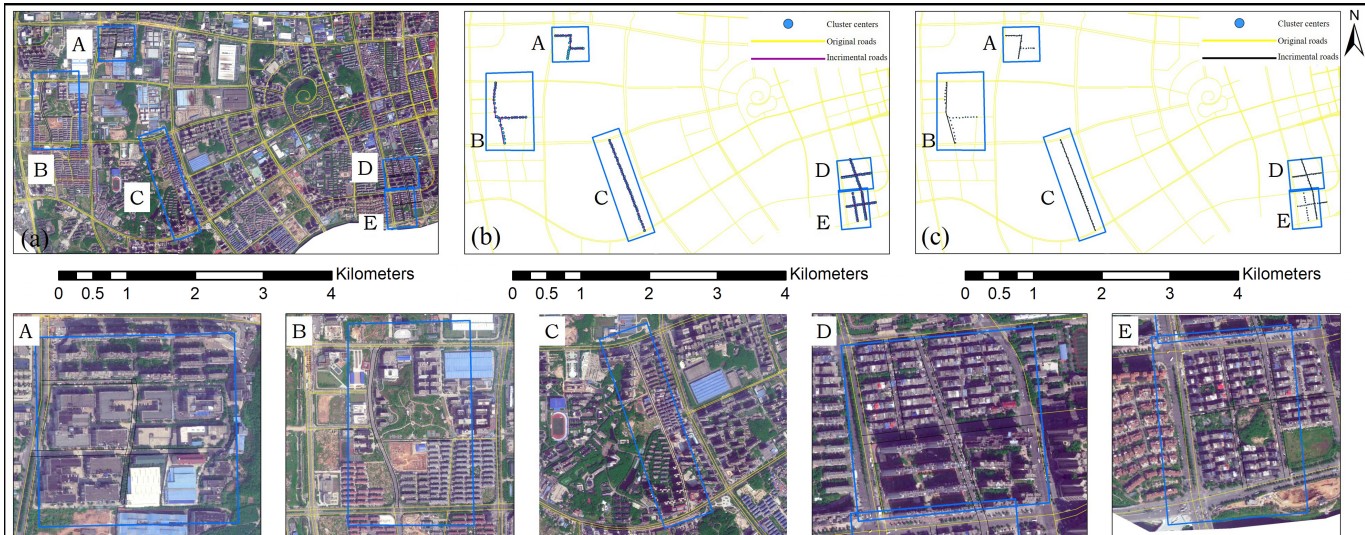

**Figure 12.** Updated results of the new road segments: (**a**) our method; (**b**) Deng et al.'s method (buffer size = 30 m); (**c**) Wu et al.'s method (cluster number = 200); and (**A**–**E**) partial enlarged drawings of updated results.

Table 2 shows the results of the proposed method compared with the other two methods. The manually corrected OSM road network in 2021 is considered the reference base. The proposed method obtained a precision of 85.31%, higher than 73.19% by Deng's method and 76.07% by Wu's method. The recall and F-score were 80.80% and 82.99%, respectively, which is higher than the other two methods. The results prove that the proposed composite update strategy can be very effective in updating new road segments and can accurately and comprehensively update new road segments.

**Table 2.** Precision, recall, and F-score of various methods with respect to new road segment updates.

| Method | Precision | Recall | *F-Score* |
|---|---|---|---|
| Our method | 85.31% | 80.80% | 82.99% |
| Deng et al. | 73.19% | 66.89% | 69.90% |
| Wu et al. | 76.07% | 58.94% | 66.42% |

The proposed method could effectively update the problematic road segments. Figure 13a shows the actual situation of the problematic road segments in the study area. Figure 13b demonstrates the updated results of the problematic road segments by the method proposed in this paper. Figure 13c shows the updated results of Deng's method. Figure 13d shows the updated results of Wu's method. The methods of Deng and Wu cannot extract the problematic road segments effectively, while the method proposed in this paper successfully updates the most problematic road segments.

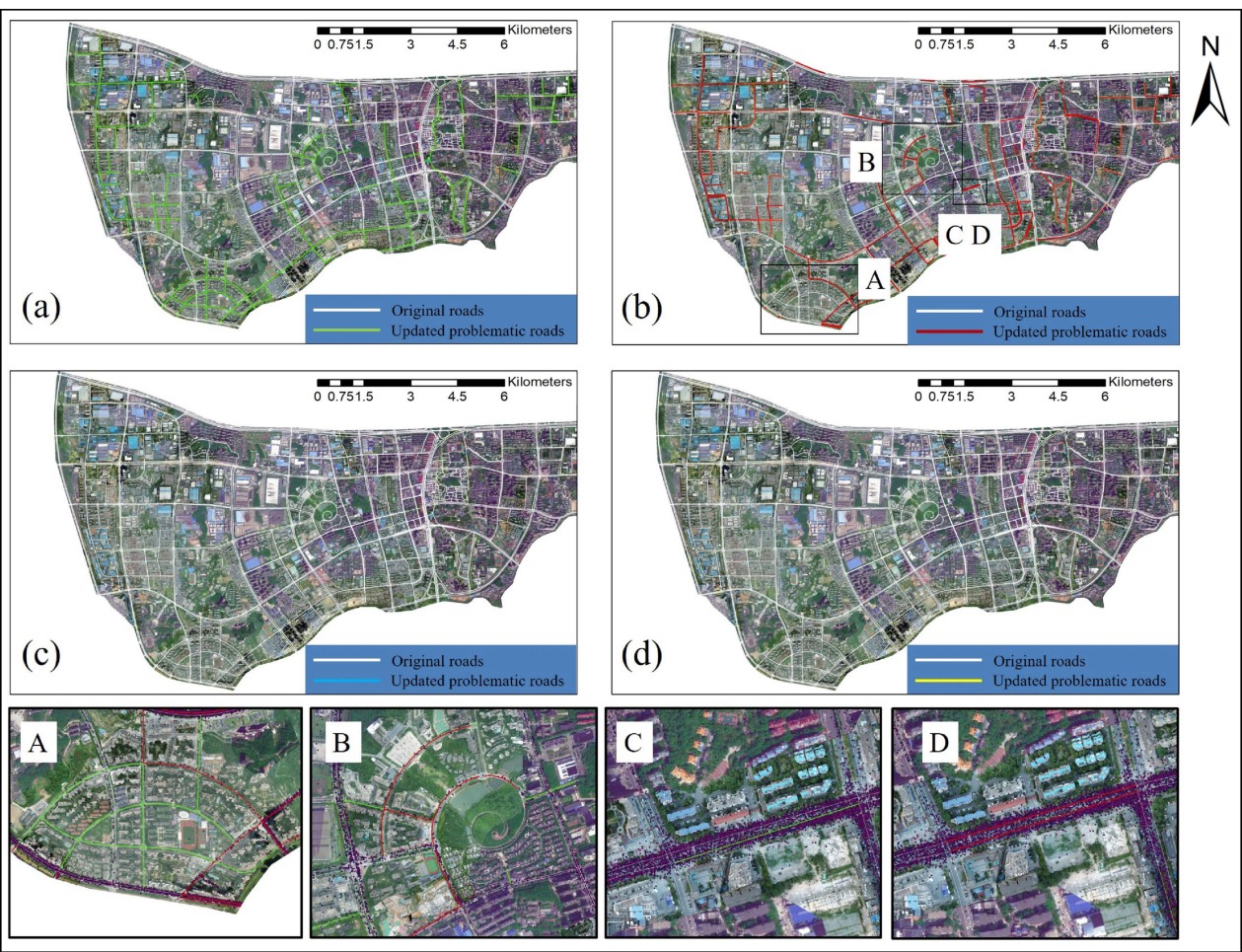

**Figure 13.** Updated results of the problematic road segments: (**a**) existing problematic road segments; (**b**) results of our method; (**c**) results of Deng et al.'s method (buffer size = 30 m); and (**d**) results of Wu et al.'s method (cluster number = 200); and (**A**–**D**) partial enlarged drawings of updated results.

Compared with the corrected OSM road network, the proposed method can achieve 79.37% accuracy, 87.72% recall, and 83.34% *F-score* for updating the problematic road segments. The results show that the proposed method has a good effect on updating the problematic road segments in the road network. The problematic road segments are shown in the partial zoom-in figure C in Figure 13, which cannot be identified by visual interpretation. However, the problematic road segments can be accurately identified by our algorithm. The updated results are shown in the partial zoom-in figure D. Compared with

the corrected road network data, as shown in the partially enlarged figureB in Figure 13, we found that a small portion of the problematic road segments was not updated. This is mainly because the road segments are very short in length, so the few points associated with the road segments are not extracted. In addition, most of the problematic segments that were not successfully extracted were because the trajectory points were not covered (as shown in the partial zoom-in figure A in Figure 13).

4.3.2. Comparison with IB Method

A total of 23 new road segments and 97 problematic road segments were extracted by the proposed method. The proposed method took a total of 31 min (including the identification of regions and the CNN-based extraction of road segment boundaries), while the IB method took 67 min (including the CNN-based extraction of road segment boundaries). The difference in time between the two methods does not appear to be significant because this study was conducted in an urban area with a dense road network. The time of the CNN-based method for pixel classification is closely related to the image size. The density of road networks in non-urban areas is about 1/20 to 1/50 of that in urban areas. The traditional IB method will take more time in other areas, and the advantage of the proposed method will be more obvious. In addition, the characteristics of the traditional method are not destined to be updated efficiently for the problematic road segments with wrong direction markers. Therefore, combining vehicle trajectories with UAV images has higher efficiency than the traditional CNN image extraction method.

**5. Conclusions and Outlook**

Modern city development requires high-speed and high-accuracy road network update methods. TB road network update methods are known for their rapid speed but poor accuracy, while IB road network update methods are known for their excellent accuracy but poor speed. Combining the advantages of these two methods, this paper proposed an incremental update method that combines GPS trajectories and UAV remote sensing imagery to achieve rapid and accurate updates to the road network. The method achieved more desirable results in practical application and has certain superiorities. Compared with other methods, the method makes the following three main contributions:

1.  A composite framework for road network update: The framework integrates the advantages of TB and IB methods to achieve rapid and accurate updating of road networks.
2.  Problematic road segment identification and extraction algorithm: The algorithm utilizes the topological relationships between adjacent matching points and the road network to identify and extract problematic road segments in the HMM map-matching process.
3.  Integrated UAV remote sensing imagery and deep learning techniques. The method can be applied to road network updating to automatically extract road boundaries and accelerate the speed of road network updating.

The proposed method took a practical step in road network data updating. Nevertheless, it considered only two spatial data types (GPS trajectories and UAV images) for road network updating. In the future, we will consider more elements together for road network updating in the following research.

**Author Contributions:** Conceptualization, Jianxin Qin and Tao Wu; Data curation, Wenjie Yang; Formal analysis, Wenjie Yang and Tao Wu; Funding acquisition, Tao Wu and Longgang Xiang; Investigation, Jianxin Qin and Wenjie Yang; Methodology, Jianxin Qin, Wenjie Yang, Tao Wu and Longgang Xiang; Resources, Jianxin Qin; Writing—original draft, Wenjie Yang, Tao Wu, Bin He and Longgang Xiang; Writing—review and editing, Jianxin Qin, Tao Wu and Longgang Xiang. All authors have read and agreed to the published version of the manuscript.

**Funding:** This research was supported in part by the National Natural Science Foundation of China under grant 41771474, and in part by the Open Research Fund of the State Key Laboratory of Information Engineering in Surveying, Mapping and Remote Sensing, Wuhan University, under grant 19I05.

**Acknowledgments:** This work has been funded by the State Key Laboratory of Information Engineering in Surveying, Mapping and Remote Sensing (State Key Lab-LIESMARS). The authors thank anonymous reviewers for their constructive comments, which helped improve the article.

**Conflicts of Interest:** The authors declare no conflict interest.

## Abbreviations

The following abbreviations are used in this manuscript:

| | |
|---|---|
| TB | trajectory-based |
| IB | image-based |
| GPS | global positioning system |
| HMM | Hidden Markov Model |
| UAV | unmanned aerial vehicle |
| DOM | digital orthophoto map |
| CNN | convolutional neural network |

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
