# Peer review of "Incremental Road Network Update Method with Trajectory Data and UAV Remote Sensing Imagery"

_ijgi, doi:10.3390/ijgi11100502_

Round 1
Reviewer 1 Report
Almost all the references were cited in the text using different styles (some of them were cited as surname+name of the first author, some were as surname+abbreviation of name, etc.) The authors must check and apply the rules (on https://www.mdpi.com/authors/references).
The references are not exactly clear why the authors refer to them. For example, the statement in the first paragraph of the introduction is so obvious and no one can disagree with these general examples. Therefore, the authors do not need to cite the first three references there.
Using abbreviations TB for trajectory-based and IB for image-based is better for readability since once in almost three sentences the methods' names are written repeatedly in sec-2. Also, the workflow must be reorganized. The images from UAV should be represented as input data like OSM and trajectory data.
The manuscript has to be edited by a professional English editor. There are many sentences with low meaning. The connections between the sentences should be reconsidered and re-corrected with formal academic language rules. It is very difficult to evaluate the article scientifically without solving all these problems. Also, it is better to write the whole abstract again and be checked in Academic English, as well.
Reviewer 2 Report
The paper presents a novel method of the road network update with combination of trajectory data and UAV imagery. The method combines strongest sides of both approaches: high speed of the trajectory-based and high accuracy of the image-based. The paper fits well into the scope of the IJGI journal. There are several issues that I would like to discuss with the authors.
GENERAL / METHODOLOGICAL ISSUES
-
I feel that the paper is a bit too complex to digest. I suggest that you review the manuscript and simplify the language.
-
You use the word 'swift' to characterize the high-speed process. It is unusual for scientific papers. I recommend changing it for more scientific term.
-
The lion's share of the references are from the Chinese authors. Aren't solutions to the problem of road network update developed by scientists from other nations as well?
-
It is unclear how you reconstruct the individual trajectories from a stream of point data. It is discribed in the paper (par. 3.1), but in too short manner. Please expand your explanations.
-
Two probabilities are introduced in the new roads region identification chapter. However, there is no place in the manuscript where the calculation of these probabilities is explained. The value of probability is within [0, 1] domain. Please provide the formulas for both of these probabilities and explain how they are calculated and used.
SPECIFIC ISSUES THROUGHOUT THE PAPER (page / lines, >> means replace)
(2 / 51) Designed a composite framework >> Composite framework
(2 / 53) Proposed a problematic road >> Problematic road
(2 / 58) Proposed a method to integrate >> Method to integrate
(2 / 72) trajectiory main includes — what is 'main' ?
(3 / 88) We reconstruct — who are 'we' here?
(3 / 108-109) the methods for road network extraction on aerial images include spectral, geometric and texture information... — information is not a method, please reformulate.
(4 / 152) The OpenStreetMap (OSM) road network data is undirected vector data — that is not true. The edge of a highway in OSM is considered to be bidirectional, unless oneway=yes tag is set.
(4 / 155) What is PostSQL and what it has to do with methodology? Is your methodology tied to specific software?
(5 / 156) serial number — condiser using the term 'index' instead
(5 / 160) data needed for this paper — this is unfortunate expression. In reality, you need the data to solve the task, not to write the paper.
(5 / 187-189) In the Hidden Markov Model (HMM) [41] based map matching process, the direction of the trajectory is often opposite to the direction of its corresponding problematic road segment — why is this happening? Please explain.
(5 / 192) New road segments are new roads built in the course of urban construction — aren't there new roads located outside of urban areas?
(6 / Figure 3) Updata >> Update
(7 / Algorithm 1) for i in N — since N is just a number, this is incorrect. Use 1..N instead.
(7 / Algorithm 1) S = NULL — this is unnecessary, since you overwrite S in line 3.
(7 / Algorithm 1) L = T-C — if these are sets, then \ operator should be used to indicate a complement operation: L = T \ C
(8 / 260) algorithm was the line >> algorithm is the line
(8 / 260-261) idea can be described by >> idea is represented in
(8 / 261-262) There is a confusion in terms here. The current point is termed 'pre' (previous) and the next point is termed 'now'. However, from a general logic, now should be current, and next should be next. Consider renaming the variables or reformulating your explanations.
(8 / 280) has a corresponding >> has the corresponding
(10 / 2) transmission >> transmission
(10 / 5) for i in X — since X is just a number, this is incorrect. Use 1..X instead.
(10 / 6) s[i-1] — this will be out of range if you start the cycle from the first index of array
(14 / 386) floating cars — what is 'floating'?
(17 / 465) On the surface — what surface is meant?
(17 / 484) Design a composite >> Composite
(17 / 487) Propose a problematic >> Problematic
(17 / 494) an agressive step — this is unfortunate expression. Pleas reformulate.
Reviewer 3 Report
The paper "An Incremental Update Method of Road Network with Trajectory Data and UAV Remote Sensing Imagery" is about the GPS trajectory and remote sensign for urban road network. The authors, in this paper, propose and implement an update mathod to achive rapid checking of the road network. The method is based on the Hidden Markov Model (HMM) with a Deep Learning method to update the original road networks.
The work is very interesting and well exposed. However, the introduction can be improved. Authors should better expose the potential of their method and better explain the technology already present.
Figure 6 should be larger. The writings are not legible.
Adjust the sentence identification. For example, After Figure 12 there is no space between the figure's caption and the text.
Figure 13, the letters in yellow are not very legible. Enlarge them or change color.
Likewise, the captions inside Figure 13 are too small.
For figure 10, 11, 12 and 13, insert scale to the aerial images.
The work is interesting and quite innovative.
Round 2
Reviewer 1 Report
- In the abstract, you may give some very brief info about the test data.
- All citations in the text should be written by typing the surnames. Do not use any abbreviation of the author's name in the text.
- Figure 6 looks like represents a screenshot of the software. If it is software, please cite it in an appropriate place like at the end of the figure caption.
- Figure 10 should contain one scale bar. And be sure that it is readable.
